# ReinAD: Towards Real-world Industrial Anomaly Detection with a Comprehensive Contrastive Dataset

**Xu Wang[1], Jingyuan Zhuo[1], Zhiyuan You[2], Zhiyu Tan[1], Yikuan Yu[1],**
**Siyu Wang[1], Xinyi Le[1],∗**

[1]Shanghai Jiao Tong University, [2]The Chinese University of Hong Kong
{wx0413, mui123, tttangerine, yyyykkkk1995, y_wsy09, lexinyi}@sjtu.edu.cn
zhiyuanyou@foxmail.com

## Abstract

Recent years have witnessed significant advancements in industrial anomaly detection (IAD) thanks to existing anomaly detection datasets. However, the large performance gap between these benchmarks and real industrial practice reveals critical limitations in existing datasets. We argue that the mismatch between current datasets and real industrial scenarios becomes the primary barrier to practical IAD deployment. To this end, we propose **ReinAD** dataset, a comprehensive contrastive dataset towards **Re**al-world **in**dustrial **A**nomaly **D**etection. Our dataset prioritizes three critical real-world requirements: 1) Contrast-based anomaly definition that is essential for industrial practice, 2) Fine-grained unaligned image pairs reflecting real inspections, and 3) Large-scale data from active production lines spanning multiple industrial categories. Based on our dataset, we introduce the ReinADNet. It takes both normal reference and test images as inputs, achieving anomaly detection through normal-anomaly comparison. To address the fine-grained and unaligned properties of real industrial scenes, our method integrates pyramidal similarity aggregation for comprehensive anomaly characterization and global-local feature fusion for spatial misalignment tolerance. Our method outperforms all baselines on the ReinAD dataset (*e.g.*, 64.5% *v.s.* 59.5% in 1-shot image-level AP) under all settings. Extensive experiments across several datasets demonstrate our dataset's challenging nature and our method's superior generalization. This work provides a solid foundation for practical industrial anomaly detection. Dataset and code are available at https://tocmac.github.io/ReinAD.

## 1 Introduction

Industrial anomaly detection (IAD) has made significant progress in recent years, benefiting from datasets such as MVTecAD [6], MPDD [33], BTAD [36], VisA [68], *etc.* Existing anomaly detection methods [32, 42, 57] have achieved remarkably high performance on these benchmarks. For example, PatchCore [42] has achieved an image-level AUROC higher than 99% on MVTecAD. However, these methods remain difficult to apply in real industrial scenarios [4, 37, 46, 49, 64]. This is mainly due to the gap between existing dataset and real industrial scenarios.

First, the contrastive ability is necessary for industrial anomaly detection. In real industrial scenarios, the identification of "which part is anomalous" should be initiated based on normal samples or rules. Notably, many industrial anomalies, such as "wire missing" in Fig. 1a, cannot be detected even by humans without the reference of normal samples. In contrast, many anomalies in existing datasets are defined only by common sense (*e.g.*, "capsule crack" in Fig. 1a), making them easier to be identified even without normal references. This is evidenced by the fact that even 0-shot IAD

---

∗Corresponding Author.

39th Conference on Neural Information Processing Systems (NeurIPS 2025) Track on Datasets and Benchmarks.

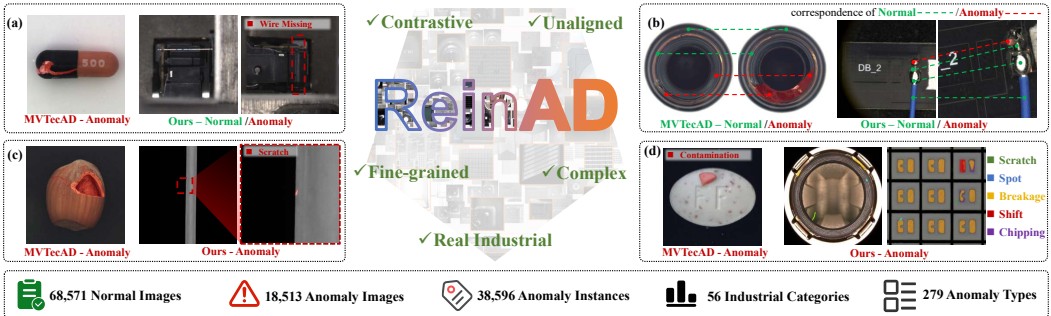

Figure 1: Illustration of our **ReinAD** dataset, a comprehensive contrastive dataset towards **Re**al-world **in**dustrial **A**nomaly **D**etection. (a) Some real anomalies (*e.g.*, "wire missing" circled by red) require contrast between normal and anomalous samples to detect. (b) Sample unalignment caused by variations in shifts, rotations, and scales in production environments. (c) Quite fine-grained anomalies (scratch) masked by red. (d) Multi-class anomalies may appear in one object.

method WinCLIP [32] has achieved a very high image-level AUROC (91.8%) on MVTecAD. Another advantage is that this contrastive ability can generalize to new categories *unseen* during training, as illustrated in prior works like InCTRL [67] and ResAD [55]. Therefore, we argue that the contrastive ability between normal and anomalous samples is crucial in industrial applications.

Second, the samples in existing datasets are mostly well aligned and the anomalies are obvious without complex categories, underestimating the challenges of actual industrial scenarios. As illustrated in Fig. 1b, all samples in the bottle category in MVTecAD dataset are well-aligned. However, images taken in real production lines exhibit variations in shifts, rotations, and scales (*e.g.*, right part in Fig. 1b) due to different production environments. Also, anomalies in existing datasets (*e.g.*, hazelnut crack in Fig. 1c) are usually obvious with a large size. In contrast, real anomalies shown in the right part of Fig. 1c can be extremely small and fine-grained. Finally, as depicted in Fig. 1d, multiple anomalies often co-occur on a single object, a scenario overlooked by existing datasets. Therefore, the difficulty of existing datasets is much lower than that of actual scenarios.

To address these challenges, we construct a large-scale dataset that matches better with real industrial demands, termed ReinAD. As illustrated in Fig. 1, Our dataset comprises four key components:

- **Contrastive capability.** We prioritize contrastive capability in sample collection. Many anomalies in our dataset can only be identified through comparison with normal samples.
- **Unaligned property.** Misalignment is common in real-world industrial imaging. Samples in our dataset capture this property through variations in shift, rotation, and scale.
- **Fine-grained anomalies.** Large quantities of anomalies in our dataset have a tiny area ratio, presenting significant challenges for anomaly detection.
- **Complex anomaly patterns.** Co-occurring anomalies are common in our dataset. This important real-world property is overlooked by many existing datasets.

Based on our ReinAD dataset, we propose ReinADNet, a model taking both normal reference and test image as inputs, identifying anomalies via comparing with normal reference. For fine-grained comparison, we propose a pyramidal cost aggregation module to compute point-wise multi-scale similarities. To contrast unaligned samples, we develop an adaptive nearest-neighbor search strategy for optimal local matching. Our method outperforms all baselines on ReinAD dataset (*e.g.*, 64.5% *v.s.* 59.5% in 1-shot image-level AP) under all settings. Cross-dataset experiments demonstrate both our dataset's challenges and our method's superior generalization.

In summary, our main contributions can be summarized as follows:

- We introduce ReinAD dataset, a novel dataset for real-world industrial anomaly detection. Our ReinAD dataset focuses on contrastive ability, containing unaligned samples and multi-class fine-grained anomalies, better reflecting real industrial scenes.
- Our comprehensive and large-scale ReinAD dataset provides a foundation for advanced anomaly detection methods. The introduced dataset contains 56 categories, 279 anomalous types, and 87,084 expert-annotated samples with anomalous segmentation masks.
- We propose ReinADNet, a generalizable anomaly detection method. ReinADNet identifies anomalies via normal-anomaly sample comparisons and handles fine-grained unaligned anomalies, achieving better results than previous baselines.

## 2 Related Works

**Anomaly detection datasets**. The evolution of anomaly detection datasets reflects incremental progress toward addressing real-world industrial challenges. Early works predominantly relied on KolektorSDD [44], a single-category dataset that constrained algorithm evaluation and development. Subsequent datasets like MTD [31], MPDD [33], and BTAD [36] expanded diversity but remained limited in scale and categorical coverage. A pivotal shift occurred with MVTec AD [6], standardizing industrial anomaly detection (IAD) research by providing 5,354 images across 15 object categories. VisA [68] further advanced this effort, scaling to 10,821 images spanning 12 objects and 15 anomaly types. However, existing datasets remain confined to narrow industrial scenarios due to their small scale and limited categories. Recent efforts like Real-IAD [46] introduced larger multi-view data, yet its reliance on artificially fabricated anomalies creates a significant domain gap in both object and anomaly realism. Meanwhile, domain-specific datasets (*e.g.*, VAD [3] for solder joints, CID [63] and CableInspect-AD [2] for cables, and 3CAD [52] for 3C components) focus on niche applications, limiting their utility for training models requiring generalizable anomaly detection capabilities across unseen industrial scenarios. These limitations underscore the urgent need for a large-scale, real-world dataset that captures the complexity and diversity of authentic industrial environments, enabling robust training and evaluation of models for generalizable anomaly detection.

**Classical anomaly detection methods**. Existing unsupervised anomaly detection methods exhibit three primary technical streams: 1) Distance-based approaches [17, 18, 24, 30, 42, 56] identify anomalies through statistical deviations in feature space; 2) Reconstruction-based methods [1, 12, 13, 28, 39, 51, 53, 54, 58, 60, 61] employ autoencoders or GANs to detect reconstruction errors; 3) Knowledge distillation-based methods [7, 9, 19, 43, 45, 47, 48] utilize teacher-student feature discrepancies. While achieving category-specific effectiveness, these methods inherently overfit to closed-set normal patterns and lack generalizable cross-category reasoning capabilities.

**Prompt-based anomaly detection methods**. Recent works leverage vision-language models (VLMs) like CLIP [41] for zero-shot detection [10, 14, 15, 23, 29, 32, 35, 40, 66], bypassing category-specific training via textual prompts. However, their performance depends critically on manual prompt design. Fixed templates show category inconsistency [11, 65], while dynamic prompts face semantic ambiguity in defining anomalies. Fundamentally, both classical and prompt-based methods focus on normality modeling rather than systematic anomaly reasoning, limiting their generalization capability.

**Generalizable anomaly detection.** Generalizable anomaly detection (GAD) seeks to develop unified detection models capable of generalizing across diverse application domains without requiring target-domain training data. The pioneering work InCTRL [67] established a baseline framework for cross-dataset anomaly classification by capturing contextual residuals between query images and normal references. While demonstrating category-generalizable detection capability, this method lacks precise anomaly localization, a critical requirement for industrial inspection scenarios. Subsequent work ResAD [55] addresses this limitation through residual feature learning with explicit normality constraints, enabling simultaneous detection and localization. Nevertheless, ResAD inherits fundamental constraints from traditional distance-based methods since its residual computation relies on global feature matching that ignores inter-image contextual relationships and intra-image neighborhood dependencies, thereby limiting its adaptability to complex anomaly patterns.

## 3 ReinAD Dataset

### 3.1 Dataset Construction

**Data collection.** Our data originates from multi-year accumulations in real industrial scenarios such as 3C electronics, mechanical components, consumer goods, *etc.* To address practical inspection needs, we develop customized optical solutions tailored for different workpieces and anomaly types (*e.g.* low-angle ring light for scratches and multi-zone light for dents), ensuring comprehensive coverage across diverse scenarios and production lines. Technicians then define anomaly criteria based on actual quality requirements and industrial SOP standards. During production, large quantities of both normal and anomaly samples are automatically captured, and subsequently labeled by annotators.

**Data annotation.** As illustrated in Fig. 2a, we design a human-in-the-loop semi-automated annotation pipeline. First, annotators manually label a small subset of samples according to predefined anomaly criteria. These annotated samples then serve as an initial training set for a segmentation model [25, 50]. The trained model subsequently generates preliminary annotations for the remaining unlabeled data.

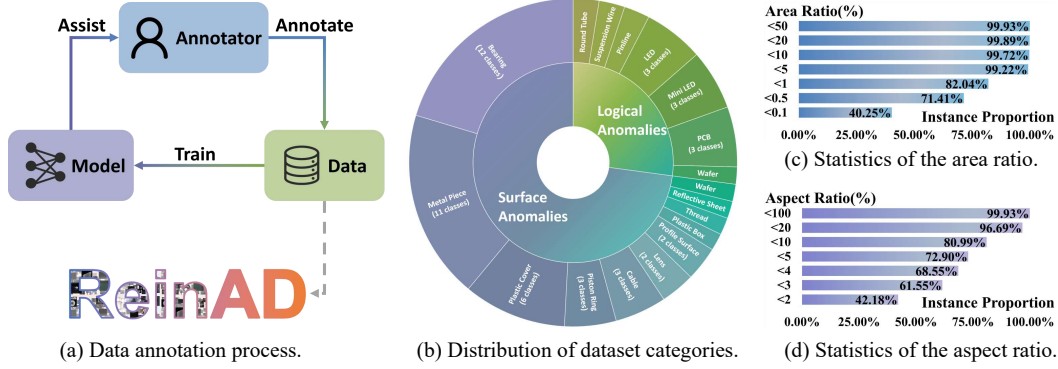

(a) Data annotation process.  (b) Distribution of dataset categories.  (c) Statistics of the area ratio.

(d) Statistics of the aspect ratio.

Figure 2: **Data annotation process and statistics** of our ReinAD dataset. (a) Data annotation process. (b) Distribution of dataset categories. (c) Statistics of the anomaly area ratio of the anomaly images. (d) Statistics of the aspect ratio of the anomaly area's minimum bounding box.

Table 1: **Comparison with existing popular anomaly detection datasets**. "%AR<0.1" denotes the percentage of samples in which the ratio of anomalous area is less than 0.1%. Missing values (*i.e.,* "-") indicate data unavailable up to submission.

| Dataset | Time | Class | Anomaly Types | Image Number | | | Anomaly Source | %AR<0.1 |
|---|---|---|---|---|---|---|---|---|
| | | | | Normal | Anomaly | Total | | |
| KSDD [44] | 2019 | 1 | 1 | 347 | 52 | 399 | Real | 8.47 |
| MVTecAD [6] | 2019 | 15 | 73 | 4,096 | 1,258 | 5,354 | Human-crafted | 1.01 |
| MTD [31] | 2020 | 1 | 6 | 952 | 392 | 1,344 | Real | 7.38 |
| KSDD2 [8] | 2021 | 1 | 5 | 2,979 | 356 | 3,335 | Real | 2.04 |
| MPDD [33] | 2021 | 6 | – | 1,064 | 282 | 1,346 | Real | 11.93 |
| BTAD [36] | 2021 | 3 | 9 | 2540 | 290 | 2,830 | Real | 6.09 |
| VisA [68] | 2022 | 12 | 75 | 9,621 | 1,200 | 10,821 | Human-crafted | 31.28 |
| MIAD [5] | 2023 | 7 | 14 | 87,500 | 17,500 | 105,000 | Virtual | 20.64 |
| Real-IAD [46] | 2024 | 30 | 131 | 99,721 | 51,329 | 151,050 | Human-crafted | – |
| VAD [3] | 2024 | 1 | 21 | 3,000 | 2,000 | 5,000 | Real | – |
| CID [63] | 2024 | 1 | 6 | 4,060 | 233 | 4,293 | Real & Synthetic | – |
| CableInspect-AD [2] | 2024 | 3 | 7 | 2,159 | 2,639 | 4,798 | Real | 0.92 |
| 3CAD [52] | 2025 | 8 | 47 | 15,577 | 11,462 | 27,039 | Real | 28.65 |
| MVTecAD-2 [26] | 2025 | 8 | 20 | 4,705 | 3,299 | 8,004 | Real | 33.65 |
| Ours | 2025 | 56 | 279 | 68,571 | 18,513 | 87,084 | Real | 40.25 |

Next, human annotators refine these annotations to produce the final high-quality ground truth. Importantly, the newly annotated data are iteratively used to retrain and improve the model. This creates a positive feedback loop that progressively enhances the model's pre-annotation accuracy. Despite this optimized semi-automated approach, pixel-level annotations for our dataset remain labor-intensive. The entire annotation process requires about 600 person-hours of expert-level effort. Quantitative details on annotation quality improvement with the human-in-the-loop annotation pipeline are available in the supplementary material.

### 3.2 Dataset Description

**Statistics.** The statistics in Fig. 2b-d demonstrate the remarkable diversity of our dataset. Fig. 2b presents the category distribution of our dataset. The anomaly types can be broadly categorized into surface anomalies and logical anomalies. Our dataset encompasses 19 industrial categories, including daily necessities, 3C components, *etc.* Each category contains one or multiple distinct products. This diversity enhances our dataset's broad applicability. As shown in Fig. 2c, our dataset contains both large-scale and small-scale anomaly regions. Fig. 2d displays the aspect ratio distribution of anomaly areas' minimum bounding boxes, revealing diverse morphological characteristics of anomalies. Both the anomaly area proportions and aspect variations indicate our dataset's high difficulty level. This is further corroborated by the experimental results in Tab. 2. We adapt a cross-category split between training and test sets to evaluate the model's generalization capability. The categories in training and test sets are completely distinct. They are randomly split while maintaining the same proportion of surface defects and logical defects.

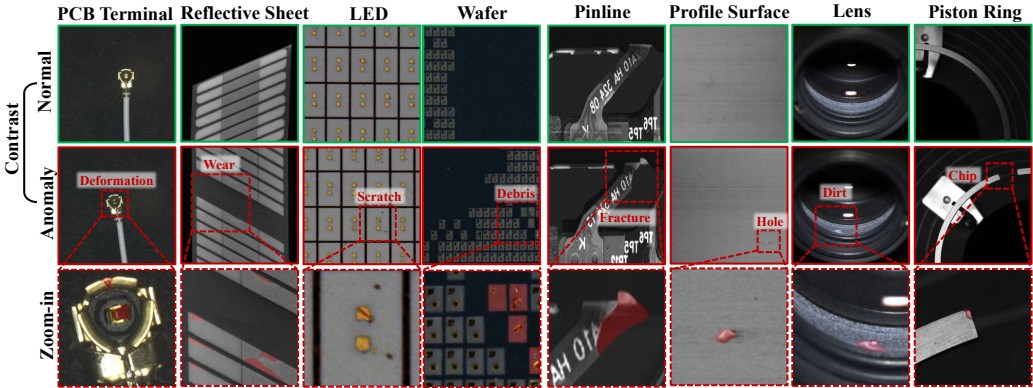

Figure 3: **Visualization of ReinAD dataset**. Samples are organized into three rows: normal images with green borders (top); anomaly images with red borders (middle); and Zoom-in patches of images in the middle row (bottom). The top black texts indicates object categories, and the red texts represents anomaly types. Additional visualizations are available in the supplementary material.

**Comparison with popular datasets.** As shown in Tab. 1, our ReinAD exhibits four key advantages over existing anomaly detection datasets. First, our dataset contains a substantial number of fine-grained anomalies, reflecting the real challenges in industrial inspection scenarios. In our dataset, samples with anomaly area ratios below 0.1% account for over 40% of the total, surpassing all other datasets listed in the table. The second-highest ratio is only 33.65% for MVTecAD-2 [26], while popular datasets like MVTec AD [6], BTAD [36] and MPDD [33] show significantly lower proportions at just 1.01%, 6.09% and 11.93% respectively. Such subtle anomalies are actually common in real industrial settings, yet current datasets notably oversimplify this critical aspect. Second, our data are entirely sourced from real industrial scenarios. All anomalies in our dataset occurred naturally during manufacturing processes. This ensures authentic representation of industrial production. In contrast, widely used datasets such as MVTec AD [6], VisA [68], and Real-IAD [46] rely on human-crafted anomalies. Such artificial anomalies exhibit significant gaps compared to real-world cases. These gaps manifest in both anomaly feature granularity and diversity of anomaly types. Third, our ReinAD serves as a comprehensive industrial dataset, offering significantly more diverse object classes and anomaly types than existing datasets. Recent datasets, such as VAD [68], CID [63], CableInspect-AD [2], and 3CAD [52], focus on specific applications (*e.g.*, solder joints, cables, or 3C components). This limits their utility for training models requiring generalizable capabilities across unseen industrial scenarios. Fourth, our dataset surpasses most datasets (except MIAD [5] and Real-IAD [46]) in data scale. Notably, MIAD is a virtual simulation dataset, and anomalies of Real-IAD are human-crafted. To our best knowledge, our dataset represents the largest real-world industrial anomaly detection dataset.

**Property analysis.** As illustrated in Fig. 1 and Fig. 3, our dataset exhibits four key characteristics. 1) *Contrastive requirement:* Many anomalies in current datasets can be simply detected without normal reference. However, real industrial anomalies (*e.g.* the PCB terminal deformation in Fig. 3) can only be identified through comparison with normal samples. 2) *Unaligned property:* Real-world industrial imaging often involves imperfect alignment. Samples in our dataset capture this through variations in shift, rotation, and scale. In Fig. 3, the wear anomalies on the reflective sheet demonstrate this characteristic. 3) *Fine-grained Anomalies:* Our dataset contains subtle anomalies in industrial settings, exemplified by the LED scratch in Fig. 3. Quantitative analysis in Fig. 2c reveals that over 40% of anomalies in our dataset have a area ratio below 0.1%, presenting significant detection challenges. 4) *Complex anomaly patterns:* Co-occurring anomalies (*e.g.* multiple wafer debris anomalies in Fig. 3) are common in our dataset. This important property is overlooked by many current datasets.

## 4 ReinADNet Method

**Problem statement.** Our objective is to achieve fine-grained, general anomaly detection. Under the contrastive paradigm, the model must jointly learn normal and anomalous patterns and transfer this discriminative ability to novel categories. To emulate such scenarios, we use a source dataset $D_{src}$ for training, where each subclass comprises normal samples $I_n$, anomalous samples $I_q$ and corresponding masks $M$. In training, it randomly samples normal–anomalous or normal–normal

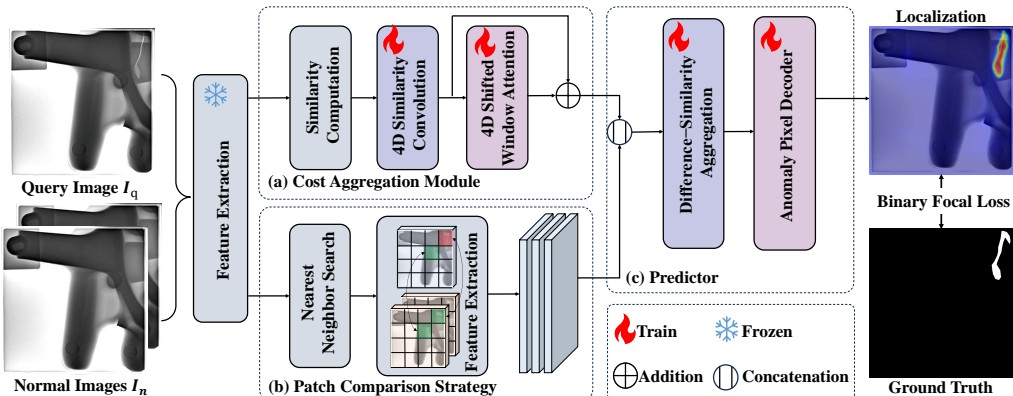

Figure 4: **Framework of our ReinADNet**. Given a query image and a set of reference images as input, a pretrained network extracts multi-scale features. The Cost Aggregation Module captures global point-to-point similarity between $I_q$ and $I_n$, while the Patch Comparison Module captures local discrepancies. The predictor subsequently aggregates these discrepancy and similarity features to generate precise pixel-wise anomaly predictions. Additional information is available in the supplementary material.

pairs across all classes, supervised by the ground-truth masks. In testing, a structurally similar target dataset $D_{tgt}$ containing unseen categories is used to evaluate the model's generalization.

**Overview of our approach.** As shown in Fig. 4, we extract multi-level features from the query image $I_q$ and a set of normal images $I_n$, forming multiple feature pairs. The cost aggregation module computes and refines the global similarity for each pair, enabling fine-grained contrast. The patch comparison strategy uses prototype learning to detect anomalies at a local scale, effectively addressing the misalignment problem. Finally, the predictor combines similarity and difference cues and further integrates each pixel's neighborhood context to decode and output the anomaly heatmap.

**Cost aggregation module.** To address global semantic shifts, we adopt insights from relevant research [16, 27] on semantic matching tasks, enabling the model to directly learn feature-to-feature similarity. First, we compute multi-level similarity between query features $f_q^l$ and normal features $f_n^l$ across $L$ hierarchical levels. Initial cost maps $C^l$ are derived via cosine similarity, where $i$ and $j$ represent the 2D spatial positions of $f_q^l$ and $f_n^l$:

$$C^l(i,j) = \frac{f_q^l(i) \cdot f_n^l(j)}{\|f_q^l(i)\|\|f_n^l(j)\|}. \tag{1}$$

Stacked cost maps $C \in \mathbb{R}^{h_q \times w_q \times h_n \times w_n \times L}$ undergo volumetric processing. A 4D CNN extracts multi-level features, followed by a 4D Swin Transformer for coarse-to-fine refinement:

$$M^l = \text{Conv4d}(C^l), \quad A^l = \text{Swin4d}(M^l), \quad A^{l-1} = \text{Swin4d}(M^l + \text{up}(A^l)). \tag{2}$$

Final feature $A \in \mathbb{R}^{h_q \times w_q \times C}$ is obtained.

**Patch-level comparison strategy.** Aligned with the paradigm of prototype learning, we propose a multi-scale patch comparison strategy. For each position $(i,j)$ in query feature $f_q$, we compute cosine distances to all patches in normal feature $f_n$, identify the closest prototype $f_{close}$, and derive local discrepancy features $f_{dist}$ as:

$$f_{close} = f_n\left(\arg\min\left(1 - \frac{f_q \cdot f_n}{\|f_q\|\|f_n\|}\right)\right), \quad f_{dist}(i,j) = f_{close}(i,j) - f_q(i,j). \tag{3}$$

Normal patches exhibit minimal $f_{dist}$, while anomalies yield larger mismatches. By deploying this module across multiple network layers, we capture scale-adaptive discrepancy features.

**Predictor.** A Swin Transformer-based module fuses the aggregated similarity features $A$ with multi-scale discrepancy features $f_{dist}$:

$$f_{fussion} = \text{Swin2d}(A \oplus f_{dist}), \quad m = \text{Conv2d}(f_{fussion}), \tag{4}$$

where $\oplus$ denotes concatenation. Predictor integrates global-local context, and generates anomaly heatmaps, with maximum anomaly score as image-level output.

**Training.** The image encoder remains frozen. Using one normal sample per class as reference, we train with normal/anomaly query pairs. Focal loss addresses class imbalance:

$$\mathcal{L} = \frac{1}{N} \sum_{x \in D_{src}} \mathcal{L}(S(x), G(x)), \tag{5}$$

where $S(x)$ is the predicted heatmap and $G(x)$ the ground truth.

**Inference.** For a test image, we compare it against reference normal samples to generate a patch-level heatmap. The maximum heatmap value determines the image-level anomaly score.

## 5 Experiments

### 5.1 Experimental Setup

**Datasets and metrics.** To assess both our dataset's challenge and our method's generalization capability, we conduct comprehensive experiments across our ReinAD dataset and several popular datasets. These datasets include MVTecAD [6], VisA [68], BTAD [36], and MPDD [33]. Previous works typically rely solely on AUROC (Area Under the Receiver Operating Characteristic Curve) as an evaluation metric. However, in anomaly detection tasks, there exists a significant class imbalance between anomalous and normal pixels, with anomalous regions accounting for only a small fraction of the total. Consequently, AUROC fails to effectively reflect model performance when influenced by numerous false positives. To address this limitation, we further incorporate image-level and pixel-level AP (Area Under the Precision-Recall Curve) and $F_1$ max scores into our evaluation for a more comprehensive assessment.

**Implementation details.** During both training and testing phases, all images are resized to $512 \times 512$ pixels and center-cropped. Following common practices in previous literature, we select WideResNet50 [59] as the feature extractor. With network parameters frozen, we utilize the outputs from all blocks of layers 2 to 4 to compute global similarity features, and we select the outputs of the final block from each of layers 1 to 3 for nearest-neighbor feature searches. We employ the Adam [34] optimizer to update network parameters, setting the learning rate to $1 \times 10^{-5}$ and weight decay to $1 \times 10^{-4}$. The total number of training epochs is set to 100, with a batch size of 4 and a random seed of 42. Similar to the training methodology of ResAD [55], we randomly select reference samples for each input image during training to enhance feature diversity. All experiments are conducted using a single NVIDIA RTX 4090 GPU.

**Competing methods.** Among traditional anomaly detection approaches, we select several classical full-shot methods and adapt them to few-shot settings, including SPADE [17], PaDiM [18], and PatchCore [42]. Additionally, we compare our approach with prompt-based methods, such as WinCLIP [32] and InCTRL [67]. Furthermore, we also include ResAD [55] and it shares a similar contrastive learning strategy with our method. Except for WinCLIP [32] and InCTRL [67] employing pretrained ViT-B-16 [21] as the backbone, all other methods utilize WideResNet50 [59] as the backbone with parameters frozen during the training phase. To ensure a fair comparison, we guarantee that all methods used the same normal samples during the testing phase.

### 5.2 Main Results

**Challenges of our ReinAD dataset.** Tab. 2 highlights the distinctive challenges of our ReinAD dataset compared to existing datasets. We first train all baselines on MVTec AD [6], then conduct 1-shot evaluation across VisA [68], BTAD [36], MPDD [33], and our ReinAD dataset. Notably, models evaluated on MVTec AD [6] are trained on VisA [68]. Experimental results reveal two key observations: (1) State-of-the-art methods have achieved strong performance on existing benchmarks: 93.7% Image-AUROC / 93.6% Pixel-AUROC on MVTec AD [6], 86.5% Image-AUROC / 95.5% Pixel-AUROC on VisA [68], and 92.3% Image-AUROC / 96.4% Pixel-AUROC on BTAD [36]. (2) However, the same methods suffer significantly reduced performance on the ReinAD dataset, with the best approach achieving only 69.0% Image-AUROC and 86.7% Pixel-AUROC. The substantial performance drop reveals that current datasets may oversimplify industrial scenarios. In contrast, our dataset contains unaligned samples and multi-class fine-grained anomalies, better matching real industrial scenes. Thus, the ReinAD dataset provides a more rigorous benchmark that encourages development of anomaly detection methods capable of handling real industrial challenges.

Table 2: **Anomaly detection and localization results** under 1-shot setting. All models are trained on MVTecAD datasets then tested on multiple datasets. Metrics are AUROC / AP / $F_1$ max. The best and second-best results are **bold** and underlined, respectively.

| | Datasets | Classical AD Methods | | | Prompt-based AD Methods | | Compare-based Methods | |
| --- | --- | --- | --- | --- | --- | --- | --- | --- |
| | | SPADE [17] | PaDiM [18] | PatchCore [42] (CVPR2022) | WinCLIP [32] (CVPR2023) | InCTRL [67] (CVPR2024) | ResAD [55] (NIPS2024) | ReinADNet (Ours) |
| Image-level | MVTecAD [6] | 72.2/86.6/87.5 | 74.5/86.5/88.7 | 82.6/91.9/91.7 | **93.7/96.9/94.5** | 88.5/93.8/91.5 | 84.3/92.7/90.7 | 85.6/93.1/89.8 |
| | VisA [68] | 73.0/77.5/78.4 | 53.2/60.4/73.8 | 74.4/78.4/78.9 | 79.9/81.8/81.3 | 75.9/78.7/78.1 | 80.3/83.8/80.4 | **86.5/89.7/84.5** |
| | BTAD [36] | 86.9/93.9/90.7 | 87.5/81.8/80.2 | 87.8/85.5/80.8 | 84.8/85.9/80.8 | 92.3/93.3/88.2 | 88.3/91.1/86.0 | 92.2/**97.8/94.8** |
| | MPDD [33] | 57.4/66.3/75.8 | 50.0/61.3/74.9 | 56.4/63.0/77.0 | **68.3/72.2/80.6** | 66.0/71.7/78.8 | 65.6/68.1/79.7 | 67.7/71.7/78.0 |
| | ReinAD (Ours) | 59.7/48.6/58.9 | 55.9/48.8/58.3 | 60.3/52.4/61.4 | **68.7**/59.5/62.6 | 59.2/53.0/61.0 | 64.9/55.0/62.5 | 68.0/**59.7/64.9** |
| Pixel-level | MVTecAD [6] | 90.5/34.2/39.0 | 88.8/32.3/37.5 | 92.1/**44.2/48.0** | 93.6/38.6/42.8 | - | 93.1/43.1/46.4 | **93.6**/43.7/46.7 |
| | VisA [68] | 92.3/14.4/20.2 | 84.9/5.6/9.5 | 93.6/26.5/31.4 | 84.6/15.8/23.4 | - | **95.5**/31.2/37.2 | 94.7/**33.2/39.0** |
| | BTAD [36] | 95.6/33.5/42.5 | 94.4/29.9/37.5 | 94.0/30.0/37.0 | 95.6/43.6/49.6 | - | 95.5/41.8/46.2 | **96.4/51.7/52.1** |
| | MPDD [33] | 93.7/14.6/19.7 | 87.5/7.5/13.3 | 93.1/16.3/18.6 | 94.4/**30.3/31.8** | - | **95.3**/24.8/26.8 | 93.8/26.6/28.4 |
| | ReinAD (Ours) | **86.7**/7.1/10.5 | 74.6/2.2/5.0 | 81.9/7.7/10.3 | 85.9/7.7/13.2 | - | 86.3/8.3/13.3 | 86.3/**10.7/15.4** |

Table 3: **Anomaly detection and localization results**. All models are trained and then tested on our ReinAD dataset under 1/2/4-shot settings. Metrics are AUROC / AP / $F_1$ max. The best and second-best results are **bold** and underlined, respectively. Detailed results for each category are available in the supplementary material.

| | Setting | ClassicalAD Methods | | | Prompt-based AD Methods | | Compare-based Methods | |
| --- | --- | --- | --- | --- | --- | --- | --- | --- |
| | | SPADE [17] | PaDiM [18] | PatchCore [42] (CVPR2022) | WinCLIP [32] (CVPR2023) | InCTRL [67] (CVPR2024) | ResAD [55] (NIPS2024) | ReinADNet (Ours) |
| Image-level | 1-shot | 59.7/48.6/58.9 | 55.9/48.8/58.3 | 60.3/52.4/61.4 | 68.7/59.5/62.6 | 53.3/49.4/58.5 | 67.0/57.5/64.5 | **71.2/64.5/67.6** |
| | 2-shot | 61.3/50.1/59.1 | 57.4/49.4/58.8 | 61.6/52.5/60.7 | 70.3/59.8/63.2 | 54.0/48.9/58.7 | 70.5/61.4/65.3 | **72.0/65.1/68.0** |
| | 4-shot | 64.1/52.3/59.7 | 63.6/51.0/60.6 | 61.9/51.7/61.0 | 71.2/60.5/63.9 | 54.6/49.3/58.7 | 73.0/63.1/66.1 | **73.8/66.2/68.0** |
| Pixel-level | 1-shot | 86.7/7.1/10.5 | 74.6/2.2/5.0 | 81.9/7.7/10.3 | 85.9/7.7/13.2 | - | 89.6/10.4/15.8 | **90.2/15.6/20.4** |
| | 2-shot | 86.1/5.2/8.8 | 75.8/2.7/5.9 | 81.9/5.7/8.8 | 86.8/8.1/13.6 | - | 91.0/12.0/18.4 | **90.3/16.3/20.8** |
| | 4-shot | 87.7/6.8/11 | 83.9/3.9/8.1 | 80.0/4.7/7.9 | 87.5/9.0/14.6 | - | **91.9**/14.5/21.4 | 89.7/**16.6/22.3** |

**Generalization ablity of our ReinADNet method.** Tab. 3 validates the generalization ability of our ReinADNet method through few-shot evaluation. All methods are trained on our ReinAD training set and evaluated on our ReinAD testing set under 1/2/4-shot settings. Experimental results reveal two critical insights: (1) Since our dataset requires normal-anomaly comparisons to identify anomalies, contrastive-based approaches (ResAD [55] and our method) dominate performance rankings across almost all settings and metrics. (2) Designed for unaligned samples and multi-class fine-grained anomalies, ReinADNet outperforms all baselines under almost all settings and metrics (*e.g.* 64.5% *v.s.* 59.5% in image-level AP under 1-shot setting). Our method achieves better generalization by treating anomaly detection as a contrastive learning paradigm rather than memorizing normal patterns. This contrastive ability can generalize to novel categories. Above results demonstrate our method's strengths of contrastive representation learning and cross-sample fine-grained alignment. Such capabilities are crucial for real-world industrial inspection scenarios.

**Qualitative results.** Fig. 5 shows qualitative results on our testing set under 1-shot setting. Most state-of-the-art methods fail to generate good anomaly localization maps for new classes, due to many false positives in normal regions. However, our method effectively reduces false positives in normal regions and locate anomalies more accurately. The LED, PCB solder and thread samples highlight our method's robust feature matching for unaligned regions, where traditional methods often fail. Additionally, the plastic cover case shows our method's exceptional sensitivity to fine-grained anomalies. It can detect subtle anomalies that baseline approaches typically miss. The visual results complement our quantitative results, confirming our ReinADNet's superiority in handling both misalignment and fine-grained anomalies.

## 5.3 Ablation Studies

Tab. 4 presents the individual and combined detection performance of each module in our method. Specifically, "Search" refers to using only the Patch Comparison Strategy, where the global features output by the similarity aggregation module are removed, while the remaining structure is kept consistent with the full model. "Aggregation" denotes the use of only the Cost Aggregation Module, where the subsequent residual feature pyramid fusion module is excluded, and the global similarity

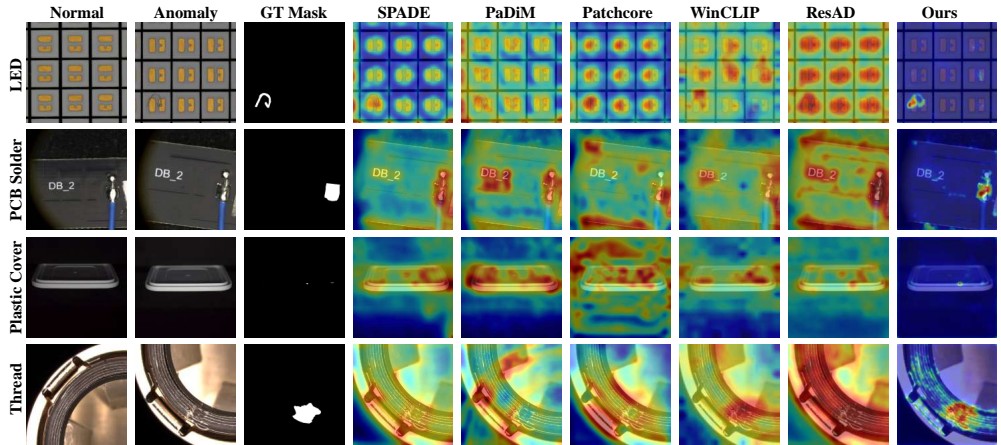

Figure 5: **Qualitative results**. More qualitative results are available in the Supplementary Material.

Table 4: **Ablation studies** of different network architectures. Metrics are AUROC / AP / $F_1$ max.

| # | Method | Image-Level | Pixel-Level |
|---|--------|-------------|-------------|
| 0 | Search | 70.0 / 61.2 / 65.7 | 88.4 / 11.9 / 17.6 |
| 1 | Aggregation | 57.9 / 50.9 / 59.3 | 86.7 / 5.5 / 9.6 |
| 2 | Aggregation+Query | 66.8 / 60.1 / 63.8 | 88.1 / 11.8 / 16.9 |
| 3 | Aggregation+Search | **71.2 / 64.5 / 67.6** | **90.2 / 15.6 / 20.4** |

Table 5: **Ablation studies** of image-level anomaly score selection strategies.

|  | I-AUROC | I-AP | I-$F_1$ max |
|---|---------|------|-------------|
| Maximum | **71.2** | **64.5** | **67.6** |
| Top 5% | 70.2 | 62.5 | 65.4 |
| Top 10% | 69.2 | 60.5 | 64.0 |
| Top 20% | 68.0 | 58.5 | 62.5 |

features are directly decoded to produce the output. "Aggregation+Query" represents a variant of the model where the features involved in the Predictor module aggregation are the original query image features rather than residual features, with the rest of the structure identical to the full model. "Aggregation+Search" denotes the complete model configuration.

**Analysis of module functionality.** As illustrated in Tab. 4, local discrepancy features derived from nearest-neighbor search effectively complement global features obtained through similarity aggregation, thereby enhancing detection performance (*i.e.* #1 *v.s.* #3), additionally, the residual features derived by feature subtraction after search mitigate the category gap and demonstrate stronger generalization capabilities than the original query features (*i.e.* #2 *v.s.* #3). The cost aggregation module consolidates global contextual information across image pairs, further refining the local differential features (*i.e.* #0 *v.s.* #3).

**Calculation of image-level anomaly scores.** The image-level anomaly scores are directly derived from the output pixel-level anomaly score maps, rather than being produced by a separately trained network. Here we compare the image-level performance on our dataset using different strategies: the maximum value of the anomaly score map, and the average of the top n% highest scores in the entire map (with n set to 5, 10, and 20). As shown in Tab. 5, the best detection performance is achieved when using the maximum anomaly value as the image-level anomaly score, and as the value of n increases, the detection performance gradually declines. This indicates that the model effectively distinguishes between normal and anomalous instances, with a significant gap between the highest anomaly score and the scores of normal regions. In contrast, introducing the top n% averaging strategy dilutes the anomaly severity and led to reduced performance.

**Contrastive-based *v.s.* zero-shot methods.** To validate the advantages of contrastive-based methods, we compare them against several zero-shot methods [11, 14, 32] on our dataset. Our contrastive setting requires simultaneous input of both normal references and query images, while most existing zero-shot methods can only accept query images as inputs. Therefore, we only input query images to evaluate zero-shot methods. The results are given in Tab. 6, the zero-shot approaches demonstrate worse performance compared to contrastive-based methods. For instance, even for our ReinAD under 1-shot setting, the advantage over WinCLIP [32] is over 5% at image-level AUROC and 12% at pixel-level AUROC. As the number of shots increases, contrastive-based methods demonstrate greater advantages over zero-shot approaches.

Table 6: **Comparison** between zero-shot methods and contrastive-based methods under 1/2/4-shot settings. Metrics are AUROC / AP / $F_1$ max.

| Shot | Method | Image-Level | Pixel-Level |
|---|---|---|---|
| 0 | APRIL-GAN [14] | 61.8/55.9/60.3 | 78.7/6.4/11.7 |
| | WinCLIP [32] | 65.5/57.0/61.0 | 77.9/2.3/5.6 |
| | AdaCLIP [11] | 64.7/58.0/61.7 | 82.1/9.1/13.7 |
| 1 | ResAD [55] | 67.0/57.5/64.5 | 89.6/10.4/15.8 |
| | ReinADNet (Ours) | 71.2/64.5/67.6 | 90.2/15.6/20.4 |
| 2 | ResAD [55] | 70.5/61.4/65.3 | 91.0/12.0/18.4 |
| | ReinADNet (Ours) | 72.0/65.1/68.0 | 90.3/16.3/20.8 |
| 4 | ResAD [55] | 73.0/63.1/66.1 | 91.9/14.5/21.4 |
| | ReinADNet (Ours) | 73.8/66.2/68.0 | 89.7/16.6/22.3 |

Table 7: **Quantitative results** of supervised defect classification methods and unsupervised anomaly detection methods on our ReinAD dataset. Here we adopt AUROC / AP / $F_1$ max as evaluation metrics.

| | Method | Image-Level | Pixel-Level |
|---|---|---|---|
| Sup. | DevNet [38] | 69.0/85.1/86.0 | - |
| | DRA [22] | 75.6/91.2/89.6 | - |
| Unsup. | SPADE [17] | 75.4/88.7/88.2 | 85.5/7.4/12.0 |
| | PaDiM [18] | 81.9/91.5/91.0 | 92.4/18.3/25.0 |
| | PatchCore [42] | 83.7/92.5/91.0 | 92.6/19.2/24.7 |
| | UniAD [57] | 74.5/88.7/88.6 | 89.5/12.6/18.8 |

## 5.4 Extended Applications of ReinAD

Beyond generalizable anomaly detection, our ReinAD dataset can be applied to extensive industrial anomaly detection tasks. First, it captures unaligned samples and multi-class fine-grained anomalies, better matching real-world complexity. Thus, it can be directly used for both one-for-one and one-for-many unsupervised anomaly detection methods. Second, as the largest real industrial dataset with pixel-level annotations, our ReinAD dataset enables backbone pre-training. Notably, the wide-used WideResNet50 [59] is pre-trained on ImageNet [20], exhibiting a critical domain gap with industrial scenarios. Therefore, a backbone pre-trained on a real industrial dataset can extract specific feature of industrial scenarios, improving the accuracy and generalization capability of IAD methods.

To demonstrating the broad applicability of our dataset, we evaluate two supervised [22, 38] and four unsupervised [17, 18, 42, 57] methods on our dataset. In these two settings, each category is split into training and test sets at an 8:1 ratio. We train the supervised models by classifying normal and anomaly samples, and then test on the same categories. The unsupervised AD methods are trained with only normal samples, and tested on the same categories. We conduct parts of experiments with the Ader [62] framework. Note that these experimental results cannot be compared with the results of few-shot methods before, since all the few-shot models are directly tested on categories unseen during training. Results in Tab. 7 demonstrate the usability of our dataset in both supervised and unsupervised settings.

## 6 Conclusion

We propose **ReinAD**, a comprehensive dataset for **Re**al-world **in**dustrial **A**nomaly **D**etection. Our dataset focuses on contrastive capability, containing unaligned samples and multi-class fine-grained anomalies. These features better match actual industrial scenarios. Based on our dataset, we introduce the ReinADNet method. Our method detects anomalies by comparing normal and anomaly samples, and can effectively identify fine-grained unaligned anomalies. Extensive experiments on ReinAD and several popular datasets demonstrate our dataset's challenge and our method's generalization ability.

**Limitation and future work.** While our ReinAD dataset offers the most diverse categories within existing industrial anomaly detection datasets, it still represents only a fraction of real industrial scenarios. Future work could extend coverage to more industrial categories, especially those with complex logical anomalies. Additionally, our method incurs higher computational costs due to its multi-scale matching approach. Thus, optimizing inference efficiency without sacrificing accuracy presents a key challenge.

## 7 Acknowledgements

This work was jointly supported by the National Natural Science Foundation of China (No.62422311), Shanghai Committee of Science and Technology, China (No.24TS1413500), the Fundamental Research Funds for the Central Universities (No.YG2025ZD07), and Shanghai Jiao Tong University 2030 Initiative.

Xu Wang and Jingyuan Zhuo contributed equally to this work.

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
