# OpenReview forum: "ReinAD: Towards Real-world Industrial Anomaly Detection with a Comprehensive Contrastive Dataset"
_NeurIPS.cc/2025/Datasets_and_Benchmarks_Track — NeurIPS 2025 Datasets and Benchmarks Track poster_

### Official Review · Reviewer_Jwkt · 2025-06-11

**Rating:** 6
**Confidence:** 5

**Summary:**

This paper introduces ReinAD, a novel industrial visual anomaly detection dataset characterized by its contrastive capability, unaligned property, fine-grained anomalies, and complex anomaly patterns. To the best of my knowledge, this is the most diverse and realistic anomaly detection dataset to date, bridging gaps left by prior datasets that are often constrained to laboratory settings or specialized niches. The dataset aligns closely with real-world industrial requirements. Furthermore, the authors propose a baseline method also named ReinAD, which emphasizes explicit comparison of query images to corresponding normal images for generalized anomaly detection. Overall, this work holds significant promise for advancing real-world applications of anomaly detection.

**Dataset Code Accessibility:**

Yes

**Ethical Considerations:**

No, there are no or only very minor ethics concerns

**Limitations Weaknesses:**

While the focus on generalization is valuable, unsupervised and supervised methods remain critical for near-term industrial adoption. Including benchmark results with state-of-the-art unsupervised methods would strengthen the paper’s impact.

As shown in Tables 2 and 3, the proposed baseline does not demonstrate performance improvements over certain existing methods (e.g., ResAD), despite incurring higher computational costs due to its 4D aggregation design. This trade-off should be critically discussed.

It would be better to change a name for the baseline method, otherwise it can cause ambiguity when refer to ReinAD.

**Strengths Contributions:**

The dataset features ~72k images spanning 59 product categories and 301 anomaly types, capturing both semantic and structural real-world anomalies. Its scale and diversity are unparalleled in existing benchmarks.

The proposed dataset comprises four key components:

**Contrastive capability.** Many anomalies require explicit comparison with normal samples for detection, a critical real-world requirement.

**Samples exhibit real-world misalignment** (shift, rotation, scale), reflecting industrial imaging challenges, in comparison to most existing datasets featuring aligned images or manually introduced misalignment.

**Fine-grained anomalies**. 40.25% of anomalies occupy less than 0.1% of the image area, addressing the subtlety of practical defects.

**Complex anomaly patterns.** Co-occurring anomalies are prevalent, a property overlooked by most existing datasets.

The proposed ReinAD baseline introduces a 4D aggregation framework, offering a novel approach to anomaly detection.

---

> ### Author Rebuttal · Authors · 2025-07-30
>
> We thank the reviewer and area chair for their efforts and appreciation of our contribution to advancing IAD application, comprehensive design of the dataset, and the novel baseline approach. We will update the manuscript as suggested. Below we address reviewer's main concerns point by point.
>
> ### **Q1: Results of supervised and unsupervised methods on ReinAD dataset.**
> Thank you for your suggestion.
> We evaluate two supervised and three unsupervised approaches on our dataset, demonstrating the broad applicability of our dataset.
> In these two settings, each category is split into training and test sets at an 8:1 ratio.
> Note that these experimental results **cannot be compared with the results in our submission**, since all models in our submission are directly tested on categories unseen during training.
>
> **Supervised methods.** We train the supervised model by classifying normal and anomaly samples, and then test on the same categories. The following results demonstrate the usability of our dataset in the supervised setting.
>
> | Methods      | Image-AUROC | Image-AP | Image-F1 max |
> |:---|:---|:---|:---|
> | DevNet [1]    |       68.95 |    85.07 |        86.03 |
> | DRA [2]    |       75.62 |    91.21|        89.56 |
>
> **Unsupervised methods.** The unsupervised AD methods are trained with only normal samples, and tested on the same categories. The results are shown in the table below.
>
> | Methods      | Image-AUROC | Image-AP | Image-F1 max | Pixel-AUROC | Pixel-AP | Pixel-F1 max |
> |:---|:---|:---|:---|:---|:---|:---|
> | SPADE  |       75.38 |    88.70 |        88.21 |       85.53 |     7.41 |        11.97 |
> | PaDiM  |       81.85 |    91.54 |        90.95 |       92.37 |    18.27 |        24.96 |
> | PatchCore |       83.69 |    92.53 |        91.02 |       92.62 |    19.20 |        24.67 |
>
> [1] Pang G, Ding C, Shen C, et al. Explainable deep few-shot anomaly detection with deviation networks[J]. arXiv preprint arXiv:2108.00462, 2021.
>
> [2] Geppert M, Larsson V, Schönberger J L, et al. Privacy preserving partial localization. In CVPR, 2022.
>
>
> ### **Q2: Performance improvements over ResAD.**
>
> We emphasize that our ReinAD method achieves significant performance improvement over ResAD.
> In particular, the advantage over ResAD is **over 3% at pixel-level AP and 4% at image-level AP** in average.
> The detailed comparison are shown in the first and second tables below, which are copied from Tab. 2 and Tab. 3 in the submission.
>
> |Datasets|Methods|Image-AUROC|Image-AP|Image-F1 max|Pixel-AUROC|Pixel-AUROC|Pixel-AUROC|
> |:---|:---|:---|:---|:---|:---|:---|:---|
> | MVTecAD        |ResAD   | 84.3        | 92.7    | 90.7     | 93.1        | 43.1    | 46.4     |
> |                |ReinAD  | 85.6 (+1.3) | 93.1 (+0.4) | 89.8 (-0.9) | 93.6 (+0.5) | 43.7 (+0.6) | 46.7 (+0.3) |
> | VisA           |ResAD   | 80.3        | 83.8    | 80.4     | 95.5        | 31.2    | 37.2     |
> |                |ReinAD  | 86.5 (+6.2) | 89.7 (+5.9) | 84.5 (+4.1) | 94.7 (-0.8) | 33.2 (+2.0) | 39.0 (+1.8) |
> | BTAD           |ResAD   | 88.3        | 91.1    | 86.0     | 95.5        | 41.8    | 46.2     |
> |                |ReinAD  | 92.2 (+3.9) | 97.8 (+6.7) | 94.8 (+8.8) | 96.4 (+0.9) | 51.7 (+9.9) | 52.1 (+5.9) |
> | MPDD           |ResAD   | 65.6        | 68.1    | 79.7     | 95.3        | 24.8    | 26.8     |
> |                |ReinAD  | 67.7 (+2.1) | 71.7 (+3.6) | 78.0 (-1.7) | 93.8 (-1.5) | 26.6 (+1.8) | 28.4 (+1.6) |
> | ReinAD Dataset |ResAD   | 64.9        | 55.0    | 62.5     | 86.3        | 8.3     | 13.3     |
> |                |ReinAD  | 68.0 (+3.1) | 59.7 (+4.7) | 64.9 (+2.4) | 86.3 (+0.0) | 10.7 (+2.4) | 15.4 (+2.1) |
> |**Avg. Improvement**|| **+3.32** | **+4.26** | **+2.54** | **-0.18** | **+3.34** | **+2.34** |
>
>
> | Settings | Methods | Image-AUROC | Image-AP | Image-F1 max | Pixel-AUROC | Pixel-AP | Pixel-F1 max |
> |:---|:---|:---|:---|:---|:---|:---|:---|
> | 1-shot | ResAD | 67.0 | 57.5 | 64.5 | 89.6 | 10.4 | 15.8 |
> | | ReinAD | 71.2 (+4.2) | 64.5 (+7.0) | 67.6 (+3.1) | 90.2 (+0.6) | 15.6 (+5.2) | 20.4 (+4.6) |
> | 2-shot | ResAD | 70.5 | 61.4 | 65.3 | 91.0 | 12.0 | 18.4 |
> | | ReinAD | 72.0 (+1.5) | 65.1 (+3.7) | 68.0 (+2.7) | 90.3 (-0.7) | 16.3 (+4.3) | 20.8 (+2.4) |
> | 4-shot | ResAD | 73.0 | 63.1 | 66.1 | 91.9 | 14.5 | 21.4 |
> | | ReinAD | 73.8 (+0.8) | 66.2 (+3.1) | 68.0 (+1.9) | 89.7 (-2.2) | 16.6 (+2.1) | 22.3 (+0.9) |
> | **Avg. Improvement** | | **+2.17** | **+4.6** | **+2.57** | **-0.77** | **+3.87** | **+2.63** |
>
> ### **Q3: Change the name of the baseline method.**
>
> Thanks for the suggestion. We will change the method's name to ReinADNet in the final version. For now, we temporarily retain this name to facilitate the reading and evaluation of other reviewers.

---

> > ### Comment · Reviewer_Jwkt · 2025-08-06
> >
> > Thank you for the detailed rebuttal. I appreciate the additional supervised/unsupervised benchmarks and the clarified performance gains; together with the planned renaming to ReinADNet these fully address my concerns. I am happy to confirm my original “Strong Accept” rating.

---

> > > ### Author Response · Authors · 2025-08-06
> > > **Thanks.**
> > >
> > > We sincerely appreciate your acknowledgment of our work. Your insightful comments have greatly enhanced the quality of our manuscript. Thanks a lot for your time and effort in reviewing this submission.

---

### Official Review · Reviewer_tJyj · 2025-06-26

**Rating:** 4
**Confidence:** 5

**Summary:**

The paper introduces ReinAD, a novel dataset and method for real-world industrial anomaly detection (IAD). The dataset addresses key limitations of existing benchmarks by focusing on contrastive capability, unaligned samples, fine-grained anomalies, and complex anomaly patterns. The proposed ReinAD method leverages normal-anomaly comparisons, pyramidal similarity aggregation, and global-local feature fusion to achieve superior performance. Experiments demonstrate the dataset's challenging nature and the method's generalization ability across multiple benchmarks.

**Dataset Code Accessibility:**

Yes

**Ethical Considerations:**

No, there are no or only very minor ethics concerns

**Final Justification:**

I keep my score.

**Limitations Weaknesses:**

1. The proposed method is only experimented in few-shot setting and lacks the results of anomaly detection experiments with full training.
2. Is the ReinAD dataset customized only for few-shot anomaly detection? How are the training and test sets divided?
3. There is a lack of full training methods the ResAD dataset, while the experimental evaluation process can be done using [1].


[1] Zhang J, He H, Gan Z, et al. Ader: A comprehensive benchmark for multi-class visual anomaly detection[J]. arXiv preprint arXiv:2406.03262, 2024, 2(3).

**Strengths Contributions:**

1. ReinAD is the largest real-world industrial anomaly detection dataset, with 59 categories, 301 anomaly types, and 71,955 expert-annotated samples. It emphasizes contrastive requirements, unaligned properties, and fine-grained anomalies, better reflecting real industrial scenarios.
2. The ReinAD method integrates contrastive learning, pyramidal similarity aggregation, and adaptive nearest-neighbor search to handle fine-grained and unaligned anomalies effectively.
3. The work bridges the gap between academic benchmarks and industrial practice, providing a foundation for practical IAD deployment.

---

> ### Author Rebuttal · Authors · 2025-07-31
>
> We thank the reviewer and area chair for their efforts and appreciation of our contribution to practical IAD research, the dataset's scale and challenging design, the generalizable baseline method, and superior performance. We will update the manuscript as suggested. Below we address reviewer's main concerns point by point.
>
> ### **Q1: Application of the method and dataset in full training tasks.**
> First, we clarify that our dataset can be used in the full-training setting. Please refer to **Q3** for results of full-training results on the ReinAD dataset.
>
> Second, our proposed model is a few-shot model, following the standard setting of prior few-shot methods [1, 2]. It takes both query images and normal references as inputs to compare their difference. We choose this setting because this comparison ability can generalize to novel categories. Therefore, our model can not be applied in the full-training setting.
>
> [1] J. Zhu and G. Pang. Toward generalist anomaly detection via in-context residual learning with few-shot sample prompts. In CVPR, 2024.
>
> [2] X. Yao, Z. Chen, C. Gao, G. Zhai, and C. Zhang. ResAD: A Simple Framework for Class Generalizable Anomaly Detection. In NeurIPS, 2024.
>
> ### **Q2: Division of training and test sets.**
> We adapt a cross-category split between training and test sets to evaluate the model's generalization capability.
> The categories in training and test sets are completely distinct.
> They are randomly split while maintaining the same proportion of surface defects and logical defects.
>
> ### **Q3: Results of full training methods on the ReinAD dataset.**
> We evaluate two supervised and three unsupervised approaches on our dataset, demonstrating its broad applicability for full-training methods.
> In these two settings, each category is split into training and test sets at an 8:1 ratio.
> Note that these experimental results **cannot be compared with the results in our submission**, since all models in our submission are directly tested on categories unseen during training.
>
> **Supervised methods.** We train the supervised model by classifying normal and anomaly samples, and then test on the same categories. The following results demonstrate the usability of our dataset in the supervised setting.
>
> | Methods      | Image-AUROC | Image-AP | Image-F1 max |
> |:---|:---|:---|:---|
> | DevNet [1]    |       68.95 |    85.07 |        86.03 |
> | DRA [2]    |       75.62 |    91.21|        89.56 |
>
> **Unsupervised methods.** The unsupervised models are trained with only normal samples, and tested on the same categories. The results are shown in the table below.
>
> | Methods      | Image-AUROC | Image-AP | Image-F1 max | Pixel-AUROC | Pixel-AP | Pixel-F1 max |
> |:---|:---|:---|:---|:---|:---|:---|
> | SPADE  |       75.38 |    88.70 |        88.21 |       85.53 |     7.41 |        11.97 |
> | PaDiM|       81.85 |    91.54 |        90.95 |       92.37 |    18.27 |        24.96 |
> | PatchCore|       83.69 |    92.53 |        91.02 |       92.62 |    19.20 |        24.67 |
>
> Thanks for the recommended framework [3]. It has significantly facilitated our evaluation process. We will cite and acknowledge this framework in our final version.
>
> [1] Pang G, Ding C, Shen C, et al. Explainable deep few-shot anomaly detection with deviation networks[J]. arXiv preprint arXiv:2108.00462, 2021.
>
> [2] Geppert M, Larsson V, Schönberger J L, et al. Privacy preserving partial localization. In CVPR, 2022.
>
> [3] Zhang J, He H, Gan Z, et al. Ader: A comprehensive benchmark for multi-class visual anomaly detection[J]. arXiv preprint arXiv:2406.03262, 2024, 2(3).

---

### Official Review · Reviewer_uFxf · 2025-07-03

**Rating:** 4
**Confidence:** 4

**Summary:**

The manuscript introduces ReinAD, a new dataset designed to address the limitations of existing industrial anomaly detection (IAD) datasets by incorporating contrastive anomaly definition, unaligned and fine-grained image pairs, and multi-class anomaly patterns. Additionally, the ReinAD method is proposed to effectively handle real-world challenges, achieving state-of-the-art results on the dataset. This work makes a meaningful contribution to advancing IAD research and provides a strong foundation for future studies. However, there are areas where the paper could be improved for better clarity, rigor, and completeness.

**Dataset Code Accessibility:**

Yes

**Ethical Considerations:**

No, there are no or only very minor ethics concerns

**Final Justification:**

Most of my concerns have been addressed, I decide to maintain the score of 4.

**Limitations Weaknesses:**

1. Provide quantitative details on how the human-in-the-loop annotation pipeline improved annotation quality over iterations.
2. Clearly define the training, validation, and testing splits for reproducibility.
3. I suggest that the authors include pseudocode or a visual illustration to clarify how this strategy addresses misalignment.
4. It is better that add annotations to highlight specific areas where the proposed method outperforms baselines.

**Strengths Contributions:**

1. ReinAD introduces a large-scale, real-world dataset with 71,955 annotated samples, covering 59 categories and 301 anomaly types. The inclusion of unaligned samples, fine-grained anomalies, and multi-class anomaly patterns enhances its practical relevance.
2. The dataset is sourced from real industrial scenarios, making it more realistic compared to existing benchmarks, which often contain artificially crafted anomalies.

---

> ### Author Rebuttal · Authors · 2025-07-31
>
> We thank the reviewer and area chair for their efforts and appreciation of our contribution to advancing IAD research, comprehensive design of the dataset, and strong performance. We will update the manuscript as suggested. Below we address reviewer's main concerns point by point.
>
> ### **Q1: Quantitative details on annotation quality improvement with the human-in-the-loop annotation pipeline.**
> Thank you for your suggestion. Our human-in-the-loop annotation pipeline significantly improves annotation accuracy and reduces human effort through iterative refinement. Below are quantitative details.
>
> ||Data (Pure human label)|Data (Mode label $\rightarrow$ human correct)|Human corrected area / GT mask area|Performance (IoU\@0.5)|
> |:---|:---|:---|:---|:---|
> |Iteration 1|5%|0%|-|**72.4%**|
> |Iteration 2|5%|10%|**35%**|**85.6%**|
> |Iteration 3|5%|50%|**12%**|**90.2%**|
> |Iteration 4|5%|95%|**4.7%**|**93.8%**|
>
> Initially, human experts annotate 5% of the data. The model trained on this dataset achieves 72.4% pixel-wise accuracy (IoU\@0.5).
> During the second iteration, the model generates pre-annotations for 10% of the data, and then human experts correct 35% of the mask area pixels. The model trained on this expanded 15% dataset then improved to 85.6% accuracy.
> With each iteration, the amount of manual correction gradually decreases, while model performance consistently improves.
> In the final iteration, human experts only need to correct 4.7% of the ground truth mask pixels, and the model achieves quite high performance (93.8% IoU\@0.5).
>
> ### **Q2: Division of training and test sets for reproducibility.**
> We will make the dataset split publicly available to ensure full reproducibility.
> We adapt a cross-category split between training and test sets to evaluate the model's generalization capability.
> The categories in training and test sets are completely distinct.
> They are randomly split while maintaining the same proportion of surface defects and logical defects.
>
> ### **Q3: Include pseudocode or a visual illustration to clarify how this strategy addresses misalignment.**
> Thank you for the suggestion. As suggested, we provide a pseudocode to clarify our method.
> To address misalignment, we developed two matching modules: one for global point-to-point matching and another for local patch-level matching, together with a matching fusion module.
> 1. **Global matching: cost aggregation.** This module is designed to establish robust global correspondences between query and reference images. It computes multi-level similarity features using cosine similarity (Eq. 1), followed by volumetric refinement (Eq. 2) via a 4D CNN and Swin Transformer. This process aggregates contextual relationships across the entire image pair, enabling tolerance to global shifts and rotations by identifying invariant correspondences.
> 2. **Local matching: patch comparison.** This module targets fine-grained local variations, such as scale changes or minor deformations. It employs a nearest-neighbor search strategy to identify optimal patch-level matches, computes a discrepancy feature (Eq. 3), and adapts to spatial irregularities. This allows the model to tolerate scale differences and localized shifts by focusing on neighborhood-level matches rather than rigid pixel alignment. In Fig. 5, our method accurately localizes anomalies in unaligned samples like thread inspections, where baselines fail due to positional offsets.
> 3. **Global and local fusion.** This fusion enables comprehensive handling of misalignment types (shifts, rotations, and scales). Ablation studies in Tab. 4 confirm that combining both modules boosts misalignment tolerance significantly (i.e., +4.4% Image-AUROC).
>
> |Algorithm: Integrated Global-Local Strategy for misalignment.|
> |---|
> |**Input:** Query image feature map $f_q \in \mathbb{R}^{H_q \times W_q \times C}$, reference normal feature map $f_n \in \mathbb{R}^{H_n \times W_ \times C}$|
> |**Output:** Anomaly heatmap $A \in \mathbb{R}^{H_q \times W_q}$|
> |**Stage 1:** The cost aggregation module for global point-to-point similarity|
> |$C = zeros(H_q, W_q, H_n, W_n)$|
> |For each position $(i, j)$ in $f_q$:|
> |&nbsp;&nbsp;For each position $(k, l)$ in $f_n$:|
> |&nbsp;&nbsp;&nbsp;&nbsp;$C(i, j, k, l) = \frac{f_q(i, j) · f_n(k, l)}{\|\|f_q(i, j)\|\| \|\|f_n(k, l)\|\|}$|
> |&nbsp;&nbsp;End|
> |End|
> |$M = Conv4d(C)$|
> |$A_{agg} = Swin4d(M)$|
> |**Stage 2:** The patch comparison module for local discrepancies|
> |$f_{dist} = zeros(H_q, W_q, C)$|
> |For each position $(i, j)$ in $f_q$:|
> |&nbsp;&nbsp;$f_{close}(i,j) = f_n (argmin_{k,l} (1-C(i,j,k,l)))$|
> |&nbsp;&nbsp;$f_{dist}(i, j) = f_{close}(i,j) - f_q(i, j)$|
> |End|
> |**Stage 3:** Fusion and final prediction|
> |$f_{fussion} = Swin2d(A_{agg} \oplus f_{dist})$|
> |$A = Conv2d(f_{fussion})$|
> |Return $A$|
>
> ### **Q4: Specific areas where the proposed method outperforms baselines.**
> Thank you for the suggestion. We expand the Tab. 3 of our submission to include the per-category 1-shot quantitative results, as the table below shows.
> Metrics are Image-AUROC / Image-AP / Image-F1 max / Pixel-AUROC / Pixel-AP / Pixel-F1 max. The best results are **bold**.
> These detailed results for each category will be included in the supplementary materials.
>
> Our ReinAD method exhibits significant advantages in handling the following challenging scenarios: anomalies requiring normal-anomaly contrast to detect (e.g., motor base and LED), misaligned samples (e.g., wafer and PCB solder), fine-grained anomalies (e.g., plastic box and plastic cover), and multi-class anomalies (e.g., bearing and lens). Please refer to the visualization results for all categories in the supplementary materials.
>
> The advantages of our method stem from both our contrastive learning paradigm and the global-local matching module. Please refer to **Q3 of reviewer uFxf and Q2 of reviewer DXJF** for detailed clarification.
>
> |Categories|Patchcore|WinCLIP|InCTRL|ResAD|ReinAD (Ours)|
> |:---|:---|:---|:---|:---|:---|
> |Bearing 1|46.6/8.0/26.7/88.5/1.6/5.8|75.0/6.8/15.1/92.7/1.1/5.3|54.5/2.5/6.6/-/-/-|80.4/31.1/45.5/98.3/10.3/25.3|**99.3**/**72.7**/**80.0**/**99.0**/**36.3**/**41.8**|
> |Cable 1|75.1/57.2/60.8/97.9/**16.9**/22.1|72.0/55.9/62.8/95.2/4.0/9.9|50.5/36.9/50.8/-/-/-|**83.7**/61.6/**73.0**/**98.5**/8.8/18.1|80.3/**65.8**/69.9/97.3/13.9/**23.8**|
> |Cable 3|87.9/80.3/80.3/98.6/21.6/30.5|87.6/78.6/78.0/99.1/17.1/27.1|58.4/42.6/57.9/-/-/-|**94.0**/86.3/**87.4**/**99.7**/41.4/47.5|90.9/**87.6**/83.2/99.3/**49.0**/**51.0**|
> |Led 1|59.3/31.0/32.6/**94.0**/**9.2**/17.0|**82.1**/**63.7**/**60.1**/85.4/6.2/12.9|72.2/45.2/42.5/-/-/-|55.0/20.4/30.4/83.9/0.3/0.7|76.6/44.8/48.1/88.6/7.8/**17.2**|
> |Led 3|73.8/29.2/42.3/92.9/0.6/1.7|71.2/39.2/40.8/91.7/4.7/12.3|60.6/27.3/30.9/-/-/-|**79.4**/44.7/**54.6**/**98.1**/**13.5**/**25.8**|77.1/**50.7**/48.8/92.5/3.0/8.4|
> |Lens 2|52.8/53.8/66.1/96.3/22.3/29.0|60.6/62.9/66.6/97.6/37.8/43.0|46.7/45.6/66.2/-/-/-|58.2/54.0/67.8/98.8/16.6/28.3|**70.5**/**72.2**/**69.6**/**99.2**/**60.0**/**58.5**|
> |Motor base 1|52.8/35.4/55.4/70.8/0.0/0.0|64.9/45.8/55.0/**83.5**/0.3/0.8|**71.1**/**51.3**/**59.1**/-/-/-|34.1/26.4/49.8/76.0/0.2/0.5|41.3/32.2/49.5/83.4/**0.4**/**1.6**|
> |Motor base 2|63.3/34.5/48.5/89.7/1.5/3.6|67.0/36.6/47.0/95.2/**9.5**/**17.0**|48.7/24.0/41.2/-/-/-|71.9/44.6/51.1/**96.2**/8.0/13.8|**72.5**/**46.1**/**52.2**/91.6/4.4/8.7|
> |Motor base 4|47.5/33.5/52.7/65.1/0.4/0.9|72.6/63.9/59.0/92.8/16.8/25.3|64.4/47.5/54.9/-/-/-|64.5/54.2/54.6/90.1/11.2/16.5|**90.3**/**89.4**/**80.9**/**98.4**/**59.5**/**60.3**|
> |Motor base 5|53.8/37.8/53.2/53.3/2.4/4.9|55.6/45.4/53.0/52.5/3.4/6.8|43.8/32.0/53.0/-/-/-|58.5/43.3/55.4/55.1/4.2/10.0|**72.2**/**64.2**/**59.8**/**57.1**/**7.0**/**12.1**|
> |Motor base 7|48.4/32.0/54.7/79.6/0.8/2.6|61.2/44.0/52.7/86.7/7.5/16.1|78.6/61.2/64.8/-/-/-|77.2/59.3/64.9/93.7/3.2/6.4|**80.6**/**71.0**/**65.7**/**97.1**/**20.3**/**27.8**|
> |Motor base 8|44.6/96.7/**98.5**/61.4/2.9/7.0|53.7/97.4/**98.5**/58.4/2.7/5.8|42.3/96.6/**98.5**/-/-/-|55.7/97.7/**98.5**/**78.3**/6.0/**11.2**|**57.5**/**98.0**/**98.5**/73.2/**6.2**/10.0|
> |PCB solder 1|**94.9**/**97.7**/**92.9**/91.0/1.8/4.2|87.2/94.5/88.0/92.9/4.7/12.2|0.0/49.3/81.3/-/-/-|88.5/95.3/88.0/**97.6**/**12.9**/**17.1**|67.9/85.0/86.7/96.4/4.8/9.1|
> |PCB solder 2|66.9/88.1/88.0/93.6/**26.7**/31.0|80.5/94.4/88.0/93.8/9.0/18.0|74.7/92.7/86.3/-/-/-|81.8/92.3/**95.6**/**98.1**/25.8/**32.5**|**93.5**/**98.0**/93.6/95.2/23.1/32.2|
> |Piston ring 1|48.3/20.8/37.0/87.7/0.0/0.0|59.9/29.2/39.2/93.5/0.1/0.2|50.8/24.2/36.5/-/-/-|**62.1**/**31.2**/**39.8**/95.9/**0.5**/**2.5**|55.1/24.5/37.8/**98.4**/0.4/1.1|
> |Piston ring 3|56.9/40.0/45.7/68.3/0.0/0.0|**64.9**/**41.8**/**49.8**/92.9/0.1/0.7|54.5/35.0/45.6/-/-/-|56.2/35.4/46.3/94.1/0.1/0.7|57.9/34.8/46.7/**97.7**/**1.3**/**4.0**|
> |Plastic box|60.7/**97.0**/**97.8**/76.1/0.0/0.0|54.9/95.5/**97.8**/90.5/0.0/0.1|12.7/90.1/97.6/-/-/-|61.4/96.5/**97.8**/85.1/0.0/0.3|**64.1**/96.7/**97.8**/**94.0**/**0.6**/**3.7**|
> |Plastic cover 1|62.9/92.5/94.8/91.2/0.0/0.2|71.4/94.8/94.8/92.6/3.7/15.6|41.8/88.1/93.7/-/-/-|72.0/94.9/**95.3**/93.9/0.8/5.1|**76.8**/**95.9**/93.7/**98.1**/**7.8**/**17.0**|
> |Profile surface 1|62.0/56.6/67.8/61.1/0.8/2.1|**66.9**/**65.9**/68.3/54.1/0.9/3.3|66.3/63.8/69.0/-/-/-|65.0/60.6/**69.1**/62.2/1.0/3.2|58.3/55.8/67.4/**73.2**/**2.7**/**7.4**|
> |Thread|40.0/29.5/50.0/54.8/0.2/0.5|**62.3**/**42.1**/**52.7**/76.1/5.0/12.1|55.0/38.5/50.0/-/-/-|43.8/32.6/50.0/**85.7**/2.3/7.9|48.5/32.8/50.0/82.5/**5.1**/**14.6**|
> |Wafer 1|78.7/80.6/71.0/96.2/58.9/63.2|**83.4**/81.1/**76.4**/82.2/30.4/35.6|65.6/63.4/65.4/-/-/-|78.5/**81.4**/69.3/**96.3**/**61.1**/**70.1**|71.0/70.0/68.9/72.9/24.2/30.2|
> |Wafer 2|50.1/20.0/34.2/94.1/0.1/0.3|56.9/29.4/34.3/91.5/4.2/**11.3**|58.9/29.6/35.0/-/-/-|51.6/22.2/34.2/95.2/1.1/3.6|**63.7**/**30.4**/**38.0**/**98.9**/**4.6**/8.0|
> |**Average**|60.3/52.4/61.4/81.9/7.7/10.3|68.7/59.5/62.6/85.9/7.7/13.2|53.3/49.4/58.5/-/-/-|67.0/57.5/64.5/89.6/10.4/15.8|**71.2**/**64.5**/**67.6**/**90.2**/**15.6**/**20.4**|

---

> > ### Comment · Reviewer_uFxf · 2025-08-05
> > **Thank you for your rebuttal**
> >
> > I have read the response, most of my concerns have been addressed.

---

> > > ### Author Response · Authors · 2025-08-06
> > > **Thanks.**
> > >
> > > We sincerely thank the reviewer for recognizing our work. The valuable suggestions have greatly improved our manuscript.

---

### Official Review · Reviewer_DXJF · 2025-07-03

**Rating:** 4
**Confidence:** 4

**Summary:**

This paper presents ReinAD, a novel fine-grained industrial anomaly detection dataset and a contrastive-learning-based method tailored to realistic inspection settings. The proposed dataset is significantly more challenging and closer to real-world use than prior benchmarks, and the method shows strong performance.

**Dataset Code Accessibility:**

Yes

**Dataset Code Comments:**

The authors provide both the dataset and code to support result reproducibility.

**Ethical Considerations:**

No, there are no or only very minor ethics concerns

**Final Justification:**

The paper presents a valuable contribution by introducing a fine-grained and large-scale industrial anomaly detection benchmark with diverse anomalies, alongside a new method (ReinAD) for improved fine-grained defect identification. The authors have addressed reviewers’ concerns in the rebuttal, including clarifications on zero-shot and full-training settings as well as additional experimental details. While the method still has the limitation of high computational cost, the experiments are comprehensive and the arguments are well-supported. Overall, I recommend Borderline Accept.

**Limitations Weaknesses:**

1. Although the dataset uses contrastive settings, it's unclear whether zero-shot methods can be evaluated under this setup. Some clarification would be helpful.
2. The method shows better generalization than prior work—please clarify the key design factors that contribute to this improvement.

**Strengths Contributions:**

1. The authors address key limitations of existing industrial anomaly detection datasets by introducing a fine-grained, large-scale, and realistic benchmark that includes subtle and diverse anomalies.
2. A new method, ReinAD, is proposed to improve the identification of fine-grained defects.
3. Extensive experiments on both ReinAD and other datasets demonstrate the challenge posed by the benchmark and the generalizability of the proposed method.
4. The paper is well-written and clearly structured, making it easy to follow.

---

> ### Author Rebuttal · Authors · 2025-07-31
>
> We thank the reviewer and area chair for their efforts and appreciation of our comprehensive design of the dataset, strong performance, extensive experiments, and clear writing. We will update the manuscript as suggested. Below we address reviewer's main concerns point by point.
>
> ### **Q1: Whether zero-shot methods can be evaluated on the dataset.**
> Zero-shot methods can be evaluated on our dataset.
> Our contrastive setting requires simultaneous input of both normal references and query images, while most existing zero-shot methods can only accept query images as inputs.
> Therefore, we only input query images to evaluate zero-shot methods.
>
> The results are given in the table below, the zero-shot approaches demonstrate worse performance compared to contrastive-based methods.
> For instance, even for our ReinAD under 1-shot setting, the advantage over WinCLIP [2] is **over 5% at image-level AUROC and 12% at pixel-level AUROC**.
> As the number of shots increases, contrastive-based methods demonstrate greater advantages over zero-shot approaches.
>
> | Settings | Methods       | Image-AUROC | Image-AP | Image-F1 max | Pixel-AUROC | Pixel-AP | Pixel-F1 max |
> |----------|---------------|-------------|----------|--------------|-------------|----------|--------------|
> | 0-shot   | APRIL-GAN [1] | 61.8        | 55.9     | 60.3         | 78.7        | 6.4      | 11.7         |
> |          | WinCLIP [2]   | 65.5        | 57.0     | 61.0         | 77.9        | 2.3      | 5.6          |
> |          | AdaCLIP [3]   | 64.7        | 58.0     | 61.7         | 82.1        | 9.1      | 13.7         |
> | 1-shot   | ResAD         | 67.0        | 57.5     | 64.5         | 89.6        | 10.4     | 15.8         |
> |          | ReinAD        | 71.2        | 64.5     | 67.6         | 90.2        | 15.6     | 20.4         |
> | 2-shot   | ResAD         | 70.5        | 61.4     | 65.3         | 91.0        | 12.0     | 18.4         |
> |          | ReinAD        | 72.0        | 65.1     | 68.0         | 90.3        | 16.3     | 20.8         |
> | 4-shot   | ResAD         | 73.0        | 63.1     | 66.1         | 91.9        | 14.5     | 21.4         |
> |          | ReinAD        | 73.8        | 66.2     | 68.0         | 89.7        | 16.6     | 22.3         |
>
> [1] X. Chen, Y. Han, and J. Zhang. April-gan: A zero-/few-shot anomaly classification and segmentation method for cvpr 2023 vand workshop challenge tracks 1&2: 1st place on zero-shot ad and 4th place on few-shot ad. arXiv preprint arXiv:2305.17382, 2023.
>
> [2] J. Jeong, Y. Zou, T. Kim, D. Zhang, A. Ravichandran, and O. Dabeer. Winclip: Zero-/few-shot anomaly classification and segmentation. In CVPR, 2023.
>
> [3] Y. Cao, J. Zhang, L. Frittoli, Y. Cheng, W. Shen, and G. Boracchi. AdaCLIP: Adapting CLIP with Hybrid Learnable Prompts for Zero-Shot Anomaly Detection. In ECCV, 2024.
>
> ### **Q2: The method's key design factors that contribute to better generalization.**
> Our method achieves better generalization by treating anomaly detection as a contrastive learning paradigm rather than memorizing normal patterns.
> Specifically, our method learns that "anomaly is the difference region betwen the test image and its normal template", and this contrastive ability can generalize to novel categories.
> In contrast, common AD methods memorize normal patterns of specific categories, and these memories can not be generalized to new categories.
>
> To cultivate the generalizable contrastive ability, we carefully design two comparison modules.
> 1. **Global matching: pyramidal cost aggregation.** This module captures point-to-point similarity for global semantic alignment. It employs hierarchical feature pairs and volumetric processing (Eq. 2) to integrate global context. This is essential for capturing semantic consistencies across misaligned samples (e.g. Fig. 1b).
> 2. **Local matching: adaptive patch comparison.** This module performs nearest-neighbor search to capture local discrepancies. By adaptively selecting the closest normal patch at each spatial position (Eq. 3), it isolates local deviations while tolerating spatial variances. This is critical for detecting fine-grained defects like micro solder excess in Fig. 1c.

---

> > ### Comment · Reviewer_DXJF · 2025-08-06
> > **Thanks for rebuttal**
> >
> > Overall, the authors have provided explanations for most of the issues. I am pleased with the author's response. I tend to maintain my initial score.

---

> > > ### Author Response · Authors · 2025-08-06
> > > **Thanks.**
> > >
> > > We are grateful for your positive assessment of our work. Your constructive feedback has significantly refined our manuscript.

---

### Note · Authors · 2025-08-12

Dear reviewers and AC,

Thank you for your time and effort in reviewing our submission. We sincerely appreciate your insightful feedback and recognition of our work. We have carefully addressed all the concerns in the rebuttal, and your valuable suggestions have significantly improved our manuscript.

Best regrads,

Authors

---

### Decision · Program_Chairs · 2025-09-18

**Decision:**

Accept (poster)

**Comment:**

After careful consideration of the reviewers’ assessments and the authors’ rebuttal, I find that the consensus leans toward acceptance. The paper makes a significant contribution to the industrial anomaly detection (IAD) community by introducing both a novel dataset and an accompanying method that directly address long-standing challenges in bridging academic research and real-world applications. Given the novelty, significance, and practical value of both the dataset and the method, I support acceptance. The ReinAD dataset provides a valuable foundation for future work, and the accompanying method demonstrates strong potential for industrial anomaly detection. The paper should be of considerable interest to the community.